# VFEM: Visual Feature Empowered Multivariate Time Series Forecasting with Cross-Modal Fusion

**Yanlong Wang**[1,2*], **Hang Yu**[3*], **Jian Xu**[1], **Fei Ma**[4], **Hongkang Zhang**[1], **Tongtong Feng**[1]
**Zijian Zhang**[5], **Shao-Lun Huang**[1†], **Danny Dongning Sun**[2†], **Xiao-Ping Zhang**[1†]

[1] *Tsinghua Shenzhen International Graduate School, Tsinghua University*  [2] *Pengcheng Laboratory*  [3] *Ant Group*
[4] *Guangdong Laboratory of Artificial Intelligence and Digital Economy (SZ)*  [5] *University of Pennsylvania*

**Reviewed on OpenReview:** *https://openreview.net/forum?id=mPhlTmYiyg*

## Abstract

Large time series foundation models often adopt channel-independent architectures to handle varying data dimensions, but this design ignores crucial cross-channel dependencies. Meanwhile, existing cross-modal methods predominantly rely on textual modalities, leaving the spatial pattern recognition capabilities of vision models underexplored for time series analysis. To address these limitations, we propose VFEM, a cross-modal forecasting model that leverages pre-trained large vision models (LVMs) to capture complex cross-variable patterns. VFEM transforms multivariate time series into visual representations, enabling LVMs to perceive spatial relationships that are not explicitly modeled by channel-independent models. Through a dual-branch architecture, visual and temporal features are independently extracted and then fused via cross-modal attention, allowing complementary information from both modalities to enhance forecasting. By freezing the LVM and training only 7.45% of the total parameters, VFEM achieves competitive performance on multiple benchmarks, offering a new perspective on multivariate time series forecasting.

## 1 Introduction

Time series forecasting has been widely applied in diverse settings such as weather, power systems, transportation, and finance (Allen et al., 2025; Wu et al., 2021b; Liu et al., 2023; Yi et al., 2023). These scenarios often involve a wide variety of temporal data, where different time series frequently exhibit intricate interrelationships. Early time series forecasting methods focus on statistical models and signal processing techniques like decomposition (Wu et al., 2021a; 2022) and frequency analysis (Zhou et al., 2022). With the development of deep learning, many studies improve prediction by fusing information across both variable and temporal dimensions (Zhang & Yan, 2023; Luo & Wang, 2024; Wang et al., 2025; 2023). The emergence of large foundation models (Ansari et al., 2024; Woo et al., 2024; Liu et al., 2024b; Goswami et al., 2024; Darlow et al., 2024) has further advanced the field, leveraging pretrained architectures to achieve strong zero-shot or few-shot forecasting performance. In addition, Mamba-based state-space architectures are also rapidly developing for time series forecasting (Jung & Kim, 2026; Karadag et al., 2026).

However, several issues remain unresolved (Tan et al., 2024; Brigato et al., 2026). Firstly, the channel-independent architecture is commonly adopted in large time series models. While this design allows the model to handle varying numbers of variables across datasets, it does not explicitly model the correlations among different variables that could enhance forecasting accuracy. Although some studies (Woo et al., 2024; Liu et al., 2024b) employ channel-dependent designs, they require full-parameter training on large-scale time series datasets, which entails substantial data requirements and computational costs. Despite this, such models may not fully capture the distinct cross-variable dependency patterns inherent to time series from

---

*Equal contribution. †Corresponding authors.

different domains, and flattening multivariate series into a univariate sequence can obscure cross-channel patterns.

Secondly, previous cross-modal fusion methods have predominantly relied on textual modalities to provide auxiliary temporal information (Jin et al., 2024; Cao et al., 2024; Sun et al., 2023; Liu et al., 2024a; 2025a). While these text-based approaches have achieved notable success, they have not fully leveraged the spatial pattern recognition capabilities inherent in vision models. Multivariate time series, when rendered as 2D images, can exhibit visual patterns—such as periodicity, lead-lag relationships, and anomalous events—that vision models are well-suited to recognize, as shown in Figure 1. Recent work, such as VisionTS (Chen et al., 2025), has begun to explore this direction. While it demonstrates the potential of vision models, it follows a channel-independent design without cross-modal fusion, leaving room for further exploration in cross-modal integration and cross-variable dependency modeling.

To address these challenges, we propose VFEM, a cross-modal forecasting framework with dual branches: a visual branch that transforms time series into 2D representations and extracts spatial patterns via a pretrained vision encoder, and a temporal branch that models sequential dependencies through self-attention. Features from both branches are concatenated and jointly processed through self-attention layers to capture cross-modal interactions. Experiments validate the effectiveness of our design and demonstrate competitive forecasting performance. This work makes three main contributions:

- We propose a cross-modal framework that transforms multivariate time series into visual representations, enabling pretrained vision models to perceive cross-channel spatial patterns. This design handles arbitrary variable dimensions while capturing complex inter-variable dependencies.

- We freeze the vision encoder and train only 7.45% of total parameters. Ablation studies validate this training strategy, and t-SNE visualization shows that the vision encoder can capture meaningful time series representations.

- Through cross-modal fusion of visual and temporal branches, VFEM achieves competitive forecasting performance on multiple benchmarks, demonstrating the effectiveness of the proposed design.

## 2 Related Work

### 2.1 Time Series Foundation Models

The emergence of foundation models has transformed time series forecasting. GPT4TS demonstrates that pretrained language model weights can transfer to temporal domains (Zhou et al., 2023). Building on this, Chronos introduces a tokenization scheme for probabilistic forecasting, achieving strong zero-shot performance (Ansari et al., 2024). Moirai and Timer propose unified frameworks pretrained on large-scale time series corpora (Woo et al., 2024; Liu et al., 2024b), while MOMENT and DAM explore diverse pretraining strategies (Goswami et al., 2024; Darlow et al., 2024). A critical architectural choice in these models is the treatment of multivariate series. Most foundation models adopt channel-independent designs, processing each variable separately to handle arbitrary input dimensions. While this enables zero-shot generalization, it does not explicitly model cross-variable dependencies that can be beneficial for forecasting. Channel-dependent alternatives like Moirai require large-scale pretraining, and may not fully capture domain-specific inter-variable patterns. Our work explores an alternative approach by leveraging visual representations that naturally encode spatial relationships between variables.

### 2.2 Cross-Modal Methods for Time Series

Cross-modal learning has emerged as a promising direction for enhancing time series models with knowledge from pretrained foundation models. The majority of existing approaches focus on the textual modality.

TimeLLM pioneers the reprogramming paradigm, transforming time series into text-like representations for frozen LLMs (Jin et al., 2024). TEMPO uses prompts to guide LLMs in decomposing time series into trend and seasonal components (Cao et al., 2024). TEST aligns time series embeddings with text prototypes

(Sun et al., 2023), while UniTime incorporates domain-specific instructions for multi-domain forecasting (Liu et al., 2024a). TimeCMA introduces explicit cross-modality alignment objectives (Liu et al., 2025a).

While text-based methods have achieved notable success, they primarily leverage LLMs' semantic understanding and reasoning capabilities. The visual modality offers complementary strengths: spatial pattern recognition, local-global feature extraction, and inherent 2D structure that naturally aligns with multivariate time series. VisionTS demonstrates this potential by reformulating forecasting as masked image reconstruction, showing that visual masked autoencoders pretrained on ImageNet can achieve competitive zero-shot performance (Chen et al., 2025).

These vision-based approaches process each channel independently and focus mainly on univariate scenarios. The potential to capture spatial relationships between multiple variables arranged in a 2D image remains unexplored. VFEM attempts to explore this direction by rendering multivariate series as images where rows represent variables and columns represent time steps, enabling vision models to perceive cross-variable patterns. Unlike existing vision-based methods, VFEM adopts a cross-modal fusion design that combines visual features with temporal representations, allowing complementary information from both modalities to enhance forecasting.

## 2.3 Multivariate Dependency Modeling

Capturing cross-variable dependencies remains an important challenge in multivariate forecasting. Crossformer introduces a two-stage attention mechanism for temporal and cross-variable modeling (Zhang & Yan, 2023). iTransformer inverts the attention paradigm, treating variables as tokens (Liu et al., 2023), while ModernTCN uses large-kernel convolutions to capture cross-channel patterns (Luo & Wang, 2024).

These approaches rely on learned weights to discover variable relationships from training data. VFEM takes a different approach by utilizing pretrained visual knowledge for cross-variable modeling. Visual patterns in rendered time series—color gradients, periodic stripes, correlation structures—share similarities with natural image patterns, which can be helpful for knowledge transfer from pretrained vision models.

Figure 1: Clear Patterns in Electricity and Traffic Time Series Visualization: Periodicity, Lead-Lag Relationships, Anomalous Events.

# 3 Methods

## 3.1 Problem Formulation

Given a multivariate time series $\mathbf{X} \in \mathbb{R}^{L \times M}$ with look-back window length $L$ and $M$ variables, where $x_{t,m}$ denotes the value of variable $m \in \{1, \ldots, M\}$ at time step $t \in \{1, \ldots, L\}$, the forecasting task aims to predict the future values $\hat{\mathbf{Y}} \in \mathbb{R}^{F \times M}$ for the next $F$ time steps. Here, channel refers to the variable dimension $M$, not image RGB channels $c$. The objective is to learn a mapping $f_\theta : \mathbb{R}^{L \times M} \to \mathbb{R}^{F \times M}$ that minimizes the mean squared error:

$$\mathcal{L}(\theta) = \frac{1}{FM} \sum_{t=1}^{F} \sum_{m=1}^{M} (\hat{y}_{t,m} - y_{t,m})^2 \tag{1}$$

Our model processes the input through two parallel modality branches—visual and temporal—before fusing them for prediction. For the visual modality, we construct the input $\mathbf{X}_{\text{vs}} \in \mathbb{R}^{M \times L \times c}$ by treating the

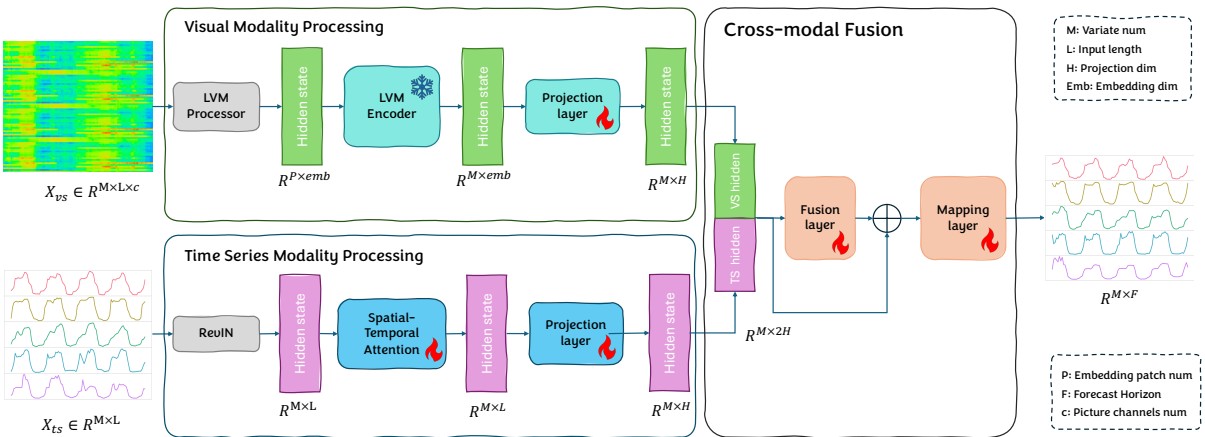

Figure 2: The overall architecture of VFEM. The model consists of two parallel branches: a visual branch that transforms time series into images and extracts features via a frozen vision encoder with projection layers, and a temporal branch that applies instance normalization followed by spatiotemporal self-attention. The outputs from both branches are concatenated and processed through a fusion encoder before the final prediction.

transposed time series matrix $\mathbf{X}^\top \in \mathbb{R}^{M \times L}$ as a single-channel intensity map and expanding it to a three-channel representation through channel replication, where $c = 3$:

$$\mathbf{X}_{\mathrm{vs}}[:,:,c] = \mathbf{X}^\top, \quad c \in \{1, 2, 3\} \tag{2}$$

In this formulation, each element of the time series directly corresponds to a pixel intensity value, where rows represent variables and columns represent time steps. This conversion process is illustrated in Figure 3. As shown in Figure 3(c-d), the color-rendered visualization highlights value variations across variables and time, while the actual input uses grayscale replication to preserve the original intensity distribution without introducing artificial color mappings.

## 3.2 Visual Analysis of Multivariate Time Series

We first examine whether multivariate time series contain visually recognizable patterns. Figure 1 shows visualizations of the ECL (Electricity) and Traffic datasets. The values across different variables exhibit distinct color variations over time, revealing clear characteristics such as daily periodic changes.

In the upper subplot displaying ECL data, different variables exhibit distinct periodic characteristics. Some variables demonstrate brief yet pronounced peak patterns, while others display superimposed daily and weekly cycles. Certain variable pairs also maintain stable lead-lag relationships, which can help capture cross-variable dependencies. The Traffic dataset visualization in the lower subplot exhibits multi-period superimposed patterns. From the zoomed-in section, distinct diurnal (day-night) traffic flow variations can be observed. Significant differences exist between weekday and weekend patterns: weekdays show pronounced morning and evening rush hours, while weekends exhibit reduced congestion. Small holidays can also be identified, showing extended periods of reduced traffic lasting more than two days.

These patterns suggest that visual representations can capture meaningful information from multivariate time series, motivating the integration of temporal and spatial dimensions in our model design.

## 3.3 Model Architecture

As depicted in Figure 2, the VFEM model comprises three key stages: (1) unimodal feature extraction through independent visual and temporal processing branches, (2) cross-modal feature fusion via concatenation, and (3) a spatiotemporal Transformer encoder followed by a prediction layer. The overall forward process can be formulated as:

$$\hat{\mathbf{Y}} = \mathcal{F}_{\mathrm{pred}} \left( \mathcal{F}_{\mathrm{fuse}} \left( [\mathbf{H}_{\mathrm{vs}}; \mathbf{H}_{\mathrm{ts}}] \right) \right) \tag{3}$$

where $[\cdot;\cdot]$ denotes concatenation along the feature dimension, $\mathbf{H}_{\text{vs}} \in \mathbb{R}^{M\times d}$ and $\mathbf{H}_{\text{ts}} \in \mathbb{R}^{M\times d}$ are the visual and temporal hidden states respectively, $\mathcal{F}_{\text{fuse}}$ represents the fusion encoder, and $\mathcal{F}_{\text{pred}}$ is the prediction layer.

### 3.4 Temporal Modality Processing

The temporal branch independently processes the raw time series through normalization and self-attention mechanisms.

**Instance Normalization.** Following the reversible instance normalization (RevIN) approach (Kim et al., 2022), we normalize the input along the temporal dimension. For each variable $m$, we compute the mean and standard deviation from the last $W$ time steps:

$$\mu_m = \frac{1}{W}\sum_{t=L-W+1}^{L} x_{t,m} \tag{4}$$

$$\sigma_m = \sqrt{\frac{1}{W}\sum_{t=L-W+1}^{L}(x_{t,m}-\mu_m)^2} \tag{5}$$

The entire sequence is then normalized using these statistics:

$$\tilde{x}_{t,m} = \frac{x_{t,m}-\mu_m}{\sigma_m+\epsilon}, \quad \forall t \in \{1,\dots,L\} \tag{6}$$

where $\epsilon$ is a small constant for numerical stability. These statistics $\mu_m$ and $\sigma_m$ are preserved and applied for denormalization after the cross-modal fusion stage, restoring the predictions to the original data scale. The normalized series $\tilde{\mathbf{X}} \in \mathbb{R}^{M\times L}$ preserves temporal patterns while removing distribution shift.

**Temporal Self-Attention.** The normalized series is processed through a spatiotemporal self-attention encoder. Given input $\mathbf{Z} \in \mathbb{R}^{M\times L}$, we first partition the temporal dimension into $N_s$ segments of length $P = L/N_s$ and rearrange the tensor to interleave spatial and temporal dimensions:

$$\mathbf{Z}' = \text{Rearrange}(\mathbf{Z}) \in \mathbb{R}^{(P\cdot M)\times N_s} \tag{7}$$

This rearrangement enables attention to capture dependencies across both variables and time segments simultaneously. The query, key, and value representations are computed via MLP-based projections with residual connections:

$$\mathbf{Q} = f_Q(\mathbf{Z}'), \quad \mathbf{K} = f_K(\mathbf{Z}'), \quad \mathbf{V} = f_V(\mathbf{Z}') \tag{8}$$

where $f_Q, f_K, f_V : \mathbb{R}^{N_s} \to \mathbb{R}^{N_s}$ are two-layer MLPs. The attention output is computed as:

$$\text{Attn}(\mathbf{Q},\mathbf{K},\mathbf{V}) = \text{softmax}\left(\frac{\mathbf{Q}\mathbf{K}^\top}{\sqrt{N_s}}\right)\mathbf{V} \tag{9}$$

The encoder applies $N_t$ layers, each containing two attention sublayers with residual connections. After processing, the output is rearranged back to the original shape:

$$\mathbf{Z}^{(l+1)} = \mathbf{Z}^{(l)} + \mathcal{E}^{(l)}(\mathbf{Z}^{(l)}), \quad l = 0,\dots,N_t-1 \tag{10}$$

with $\mathbf{Z}^{(0)} = \tilde{\mathbf{X}}$, where $\mathcal{E}^{(l)}$ denotes the $l$-th encoder layer. The output is projected to the hidden dimension:

$$\mathbf{H}_{\text{ts}} = \mathbf{Z}^{(N_t)}\mathbf{W}_{\text{ts}} \in \mathbb{R}^{M\times d} \tag{11}$$

where $d$ is the unified hidden dimension.

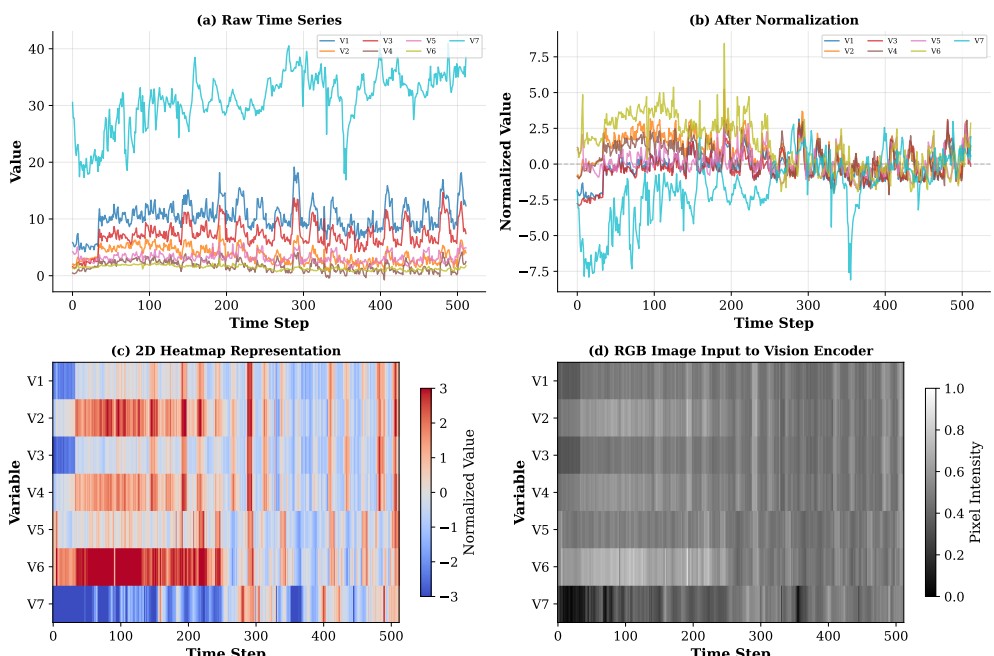

Figure 3: Visual input construction process. (a) Original multivariate time series $\mathbf{X} \in \mathbb{R}^{L \times M}$. (b) Transposed matrix $\mathbf{X}^\top \in \mathbb{R}^{M \times L}$ with variables as rows and time steps as columns. (c) Color-rendered visualization for demonstration, where value magnitudes are mapped to colors. (d) Actual input format: three-channel grayscale image $\mathbf{X}_{vs} \in \mathbb{R}^{M \times L \times c}$ with $c = 3$, created by replicating the transposed matrix across RGB channels.

## 3.5 Visual Modality Processing

The visual branch leverages a pretrained vision encoder to extract semantic representations from the constructed visual input. We employ SigLIP-2-base-NaFlex (Tschannen et al., 2025), a vision-language model that supports flexible input resolutions, as the visual backbone. All parameters of the vision encoder remain frozen during training.

**Image Encoding.** The visual input $\mathbf{X}_{vs}$ is processed by the vision encoder to obtain a global image representation:

$$\mathbf{e}_{vs} = \mathcal{V}_{enc}(\mathbf{X}_{vs}) \in \mathbb{R}^{d_v} \tag{12}$$

where $\mathcal{V}_{enc}(\cdot)$ denotes the frozen vision encoder and $d_v$ is the visual embedding dimension.

**Variable-wise Expansion.** Since the vision model produces a single global embedding capturing the holistic patterns of the entire visual representation, we replicate this embedding across all $M$ variables to enable each variable to access the shared visual context during cross-modal fusion:

$$\mathbf{E}_{vs} = \mathbf{1}_M \otimes \mathbf{e}_{vs} \in \mathbb{R}^{M \times d_v} \tag{13}$$

where $\mathbf{1}_M \in \mathbb{R}^M$ is an all-ones vector and $\otimes$ denotes the outer product.

**Feature Projection.** The expanded embedding is transformed through a learnable projection network to bridge the domain gap between pretrained visual features and time series representations. The projection includes residual connections:

$$\mathbf{H}'_{vs} = \phi(\mathbf{E}_{vs}\mathbf{W}_1)\mathbf{W}_2 + \mathbf{E}_{vs}\mathbf{W}_1 \tag{14}$$

where $\phi(\cdot)$ is the GELU activation function and $\mathbf{W}_1, \mathbf{W}_2 \in \mathbb{R}^{d_v \times d_v}$ are learnable matrices. The output is mapped to the hidden dimension:

$$\mathbf{H}_{vs} = \mathbf{H}'_{vs}\mathbf{W}_{vs} \in \mathbb{R}^{M \times d} \tag{15}$$

Table 1: Parameter distribution in the VFEM model.

| | Frozen Part | Trainable Part | | | |
| | Visual Encoder | Visual Projection | Temporal Encoder | Fusion Network | Total |
|---|---|---|---|---|---|
| **Parameters** | 375M | 1.8M | 16M | 12M | 30M |
| **Proportion** | 92.55% | 0.45% | 4.00% | 3.00% | 7.45% |

By freezing the vision encoder, the trainable parameters are reduced to only 30M, accounting for 7.45% of the total, as shown in Table 1.

### 3.6 Cross-Modal Fusion

After obtaining the visual and temporal hidden states from their respective branches, we perform cross-modal fusion by concatenating the representations:

$$\mathbf{H}_{\mathrm{f}} = [\mathbf{H}_{\mathrm{vs}}; \mathbf{H}_{\mathrm{ts}}] \in \mathbb{R}^{M \times 2d} \tag{16}$$

The concatenated features are processed through a fusion encoder that shares the same attention structure as the temporal encoder, with $N_f$ self-attention layers operating on the combined $2d$-dimensional features:

$$\mathbf{H}^{(l+1)} = \mathbf{H}^{(l)} + \mathrm{Attn}^{(l)}(\mathbf{H}^{(l)}), \quad l = 0, \ldots, N_f - 1 \tag{17}$$

with $\mathbf{H}^{(0)} = \mathbf{H}_{\mathrm{f}}$. Since the concatenated dimension contains features from both modalities, the attention mechanism models cross-modal interactions alongside cross-variable dependencies.

### 3.7 Prediction and Denormalization

The fusion encoder output is passed through a linear prediction layer:

$$\hat{\mathbf{Y}}_{\mathrm{norm}} = \mathbf{H}^{(N_f)} \mathbf{W}_{\mathrm{pred}} \in \mathbb{R}^{M \times F} \tag{18}$$

where $\mathbf{W}_{\mathrm{pred}} \in \mathbb{R}^{2d \times F}$ projects the hidden representation to the forecast horizon.

Finally, we apply reverse instance normalization to recover the predictions in the original scale:

$$\hat{y}_{t,m} = \hat{y}_{t,m}^{\mathrm{norm}} \cdot (\sigma_m + \epsilon) + \mu_m \tag{19}$$

where $\mu_m$ and $\sigma_m$ are the per-variable statistics computed during input normalization.

## 4 Experiments

### 4.1 Setup

Since the number of variates in multivariate time series varies across different datasets, and the variable dimension and temporal dimension length of the input time series are often highly imbalanced, we adopt the encoder of SigLip2-base-Naflex (Tschannen et al., 2025) as the backbone of the vision encoder, which allows arbitrary adjustment of image height and width ($d_v = 768$). During training, all parameters of the visual encoder are frozen, and only the remaining parts of VFEM are trained, with trainable parameters accounting for only 7.45% of the total parameters.

The key hyperparameters are configured as follows: input sequence length $L = 512$, prediction horizon $F \in \{96, 192, 336, 720\}$, normalization window $W = 512$, number of segments $N_s = 32$, hidden dimension $d = 1024$, number of temporal encoder layers $N_t = 1$, and number of fusion encoder layers $N_f = 1$. We use the Adam optimizer with learning rate $10^{-4}$ and train for 300 epochs with early stopping.

Table 2: Forecasting results comparison (MSE). Evaluated with prediction horizon $F \in \{96, 192, 336, 720\}$. Lower values indicate better performance. **Bold** represents the best result, and underline represents the second best.

| Models | | VFEM (Ours) | Time Series Foundation Models (Zero-Shot) | | | | Task-Specific Models (Supervised) | | | |
|---|---|---|---|---|---|---|---|---|---|---|
| | | | Chronos$_{Base}$ | Chronos$_{Large}$ | Moirai$_{Base}$ | Moirai$_{Large}$ | GPT4TS | UniTST | TimesNet | PatchTST |
| ETTh1 | 96 | **0.348**$_{\pm0.001}$ | 0.440 | 0.441 | 0.376 | 0.381 | 0.376 | 0.383 | 0.452 | 0.404 |
| | 192 | **0.383**$_{\pm0.002}$ | 0.492 | 0.502 | 0.412 | 0.434 | 0.416 | 0.434 | 0.474 | 0.454 |
| | 336 | **0.398**$_{\pm0.002}$ | 0.550 | 0.576 | 0.433 | 0.485 | 0.442 | 0.471 | 0.493 | 0.497 |
| | 720 | **0.401**$_{\pm0.003}$ | 0.882 | 0.835 | 0.447 | 0.611 | 0.477 | 0.479 | 0.560 | 0.496 |
| | Avg | **0.383**$_{\pm0.002}$ | 0.591 | 0.589 | 0.417 | 0.478 | 0.428 | 0.442 | 0.495 | 0.463 |
| ETTh2 | 96 | **0.266**$_{\pm0.002}$ | 0.308 | 0.320 | 0.294 | 0.296 | 0.285 | 0.292 | 0.340 | 0.312 |
| | 192 | **0.328**$_{\pm0.002}$ | 0.384 | 0.406 | 0.365 | 0.361 | 0.354 | 0.370 | 0.402 | 0.397 |
| | 336 | **0.357**$_{\pm0.002}$ | 0.429 | 0.492 | 0.376 | 0.390 | 0.373 | 0.382 | 0.452 | 0.435 |
| | 720 | **0.372**$_{\pm0.003}$ | 0.501 | 0.603 | 0.416 | 0.423 | 0.406 | 0.409 | 0.462 | 0.436 |
| | Avg | **0.331**$_{\pm0.002}$ | 0.406 | 0.455 | 0.363 | 0.368 | 0.355 | 0.363 | 0.414 | 0.395 |
| ETTm1 | 96 | **0.291**$_{\pm0.001}$ | 0.454 | 0.457 | 0.363 | 0.380 | 0.292 | 0.313 | 0.338 | 0.344 |
| | 192 | **0.325**$_{\pm0.002}$ | 0.567 | 0.530 | 0.388 | 0.412 | 0.332 | 0.359 | 0.371 | 0.367 |
| | 336 | **0.355**$_{\pm0.002}$ | 0.662 | 0.577 | 0.416 | 0.436 | 0.366 | 0.395 | 0.410 | 0.392 |
| | 720 | **0.406**$_{\pm0.002}$ | 0.900 | 0.660 | 0.460 | 0.462 | 0.417 | 0.449 | 0.478 | 0.464 |
| | Avg | **0.344**$_{\pm0.002}$ | 0.646 | 0.556 | 0.407 | 0.423 | 0.352 | 0.379 | 0.399 | 0.392 |
| ETTm2 | 96 | **0.166**$_{\pm0.001}$ | 0.199 | 0.197 | 0.205 | 0.211 | 0.173 | 0.178 | 0.187 | 0.177 |
| | 192 | **0.220**$_{\pm0.001}$ | 0.261 | 0.254 | 0.275 | 0.281 | 0.229 | 0.243 | 0.249 | 0.246 |
| | 336 | **0.276**$_{\pm0.002}$ | 0.326 | 0.313 | 0.329 | 0.341 | 0.286 | 0.302 | 0.321 | 0.305 |
| | 720 | **0.368**$_{\pm0.003}$ | 0.455 | 0.416 | 0.437 | 0.485 | 0.378 | 0.398 | 0.497 | 0.410 |
| | Avg | **0.258**$_{\pm0.002}$ | 0.310 | 0.295 | 0.312 | 0.330 | 0.267 | 0.280 | 0.314 | 0.285 |
| ECL | 96 | **0.128**$_{\pm0.002}$ | 0.154 | 0.152 | 0.160 | 0.153 | 0.139 | 0.139 | 0.184 | 0.186 |
| | 192 | **0.147**$_{\pm0.001}$ | 0.179 | 0.172 | 0.175 | 0.169 | 0.153 | 0.155 | 0.192 | 0.190 |
| | 336 | **0.161**$_{\pm0.001}$ | 0.214 | 0.203 | 0.187 | 0.187 | 0.169 | 0.170 | 0.200 | 0.206 |
| | 720 | **0.193**$_{\pm0.002}$ | 0.311 | 0.289 | 0.228 | 0.237 | 0.206 | 0.198 | 0.228 | 0.247 |
| | Avg | **0.157**$_{\pm0.002}$ | 0.215 | 0.204 | 0.188 | 0.187 | 0.167 | 0.166 | 0.201 | 0.207 |
| Weather | 96 | **0.153**$_{\pm0.002}$ | 0.203 | 0.194 | 0.220 | 0.199 | 0.162 | 0.156 | 0.169 | 0.177 |
| | 192 | **0.201**$_{\pm0.001}$ | 0.256 | 0.249 | 0.271 | 0.246 | 0.204 | 0.207 | 0.222 | 0.222 |
| | 336 | **0.250**$_{\pm0.002}$ | 0.314 | 0.302 | 0.286 | 0.274 | 0.254 | 0.263 | 0.290 | 0.277 |
| | 720 | **0.319**$_{\pm0.002}$ | 0.397 | 0.372 | 0.373 | 0.337 | 0.326 | 0.340 | 0.376 | 0.352 |
| | Avg | **0.231**$_{\pm0.002}$ | 0.293 | 0.279 | 0.288 | 0.264 | 0.237 | 0.242 | 0.264 | 0.257 |
| Traffic | 96 | **0.362**$_{\pm0.001}$ | 0.523 | 0.502 | 0.636 | 0.591 | 0.388 | 0.402 | 0.593 | 0.462 |
| | 192 | **0.376**$_{\pm0.002}$ | 0.570 | 0.554 | 0.702 | 0.610 | 0.407 | 0.426 | 0.596 | 0.466 |
| | 336 | **0.398**$_{\pm0.001}$ | 0.632 | 0.578 | 0.756 | 0.648 | 0.412 | 0.449 | 0.600 | 0.482 |
| | 720 | **0.429**$_{\pm0.006}$ | 0.729 | 0.646 | 0.804 | 0.712 | 0.450 | 0.489 | 0.619 | 0.514 |
| | Avg | **0.391**$_{\pm0.003}$ | 0.614 | 0.570 | 0.725 | 0.640 | 0.414 | 0.442 | 0.602 | 0.481 |

We select multiple competitive models as baselines, covering diverse approaches: large-scale foundation models (Chronos and Moirai (Ansari et al., 2024; Woo et al., 2024) in both Base and Large versions), LLM-based cross-modal methods (GPT4TS (Zhou et al., 2023)), and specialized time series architectures (UniTST (Liu et al., 2025b), TimesNet (Wu et al., 2022), PatchTST (Nie et al., 2023)).Chronos and Moirai are evaluated as zero-shot time series foundation model (TSFM) baselines, while VFEM and the remaining baselines are Task-Specific Models (Supervised), trained on downstream labels. This evaluation setup follows (Liu et al., 2025c) and aligns with the alternative fine-tuning pipeline proposed in (Shi et al., 2025), where TSFM baselines are evaluated without downstream fine-tuning, while task-specific models are tuned on downstream labels. For fair comparison, we use results from original publications when available; for Chronos and Moirai MSE/MAE, we use the zero-shot values reported in (Liu et al., 2025c), except on Traffic where we report our own zero-shot results under the same evaluation setting. All experiments follow the same evaluation protocol with identical dataset splits and metrics. Experiments were conducted on 7 datasets, including ETTh1, ETTh2, ETTm1, ETTm2, Electricity, Weather, and Traffic. To ensure the robustness of results, each experiment was run 5 times with different random seeds. We report the average test metrics with standard deviation for VFEM.

## 4.2 Results

As shown in Tables 2 and 3, the VFEM model achieved lower MSE and MAE values on 7 datasets. The visual modality also enhances the ability to recognize long-term patterns in time series, which makes the model performance degrade more slowly in long-sequence forecasting. For example, the MSE loss with the

Table 3: Forecasting results comparison (MAE). Evaluated with prediction horizon $F \in \{96, 192, 336, 720\}$. Lower values indicate better performance. **Bold** represents the best result, and underline represents the second best.

| Models | | VFEM (Ours) | Time Series Foundation Models (Zero-Shot) | | | | Task-Specific Models (Supervised) | | | |
|---|---|---|---|---|---|---|---|---|---|---|
| | | | Chronos$_{Base}$ | Chronos$_{Large}$ | Moirai$_{Base}$ | Moirai$_{Large}$ | GPT4TS | UniTST | TimesNet | PatchTST |
| ETTh1 | 96 | **0.383**$_{\pm0.001}$ | 0.393 | 0.390 | 0.392 | 0.388 | 0.397 | 0.398 | 0.463 | 0.413 |
| | 192 | **0.405**$_{\pm0.002}$ | 0.426 | 0.524 | 0.413 | 0.415 | 0.418 | 0.426 | 0.477 | 0.430 |
| | 336 | **0.414**$_{\pm0.001}$ | 0.462 | 0.467 | 0.428 | 0.445 | 0.433 | 0.445 | 0.489 | 0.462 |
| | 720 | **0.434**$_{\pm0.002}$ | 0.591 | 0.583 | 0.444 | 0.510 | 0.515 | 0.469 | 0.534 | 0.481 |
| | Avg | **0.409**$_{\pm0.002}$ | 0.468 | 0.491 | 0.419 | 0.440 | 0.441 | 0.435 | 0.491 | 0.447 |
| ETTh2 | 96 | **0.330**$_{\pm0.001}$ | 0.343 | 0.345 | **0.330** | **0.330** | 0.342 | 0.342 | 0.374 | 0.358 |
| | 192 | **0.369**$_{\pm0.002}$ | 0.392 | 0.399 | 0.375 | 0.371 | 0.389 | 0.390 | 0.414 | 0.408 |
| | 336 | **0.394**$_{\pm0.001}$ | 0.430 | 0.453 | 0.390 | 0.390 | 0.407 | 0.408 | 0.452 | 0.440 |
| | 720 | **0.413**$_{\pm0.002}$ | 0.477 | 0.511 | 0.433 | 0.418 | 0.441 | 0.431 | 0.468 | 0.449 |
| | Avg | **0.377**$_{\pm0.002}$ | 0.411 | 0.427 | 0.382 | 0.377 | 0.395 | 0.393 | 0.427 | 0.414 |
| ETTm1 | 96 | **0.343**$_{\pm0.001}$ | 0.408 | 0.403 | 0.356 | 0.361 | 0.346 | 0.352 | 0.375 | 0.373 |
| | 192 | **0.364**$_{\pm0.001}$ | 0.477 | 0.450 | 0.375 | 0.383 | 0.372 | 0.380 | 0.387 | 0.386 |
| | 336 | **0.381**$_{\pm0.002}$ | 0.525 | 0.481 | 0.392 | 0.400 | 0.394 | 0.404 | 0.411 | 0.407 |
| | 720 | **0.411**$_{\pm0.002}$ | 0.591 | 0.526 | 0.418 | 0.420 | 0.421 | 0.440 | 0.450 | 0.442 |
| | Avg | **0.375**$_{\pm0.002}$ | 0.500 | 0.465 | 0.385 | 0.391 | 0.383 | 0.394 | 0.406 | 0.402 |
| ETTm2 | 96 | **0.255**$_{\pm0.001}$ | 0.274 | 0.271 | 0.273 | 0.274 | 0.262 | 0.262 | 0.267 | 0.260 |
| | 192 | **0.292**$_{\pm0.001}$ | 0.322 | 0.314 | 0.316 | 0.318 | 0.301 | 0.304 | 0.309 | 0.305 |
| | 336 | **0.328**$_{\pm0.002}$ | 0.366 | 0.353 | 0.350 | 0.355 | 0.341 | 0.341 | 0.351 | 0.343 |
| | 720 | **0.385**$_{\pm0.002}$ | 0.439 | 0.415 | 0.411 | 0.428 | 0.401 | 0.395 | 0.403 | 0.405 |
| | Avg | **0.315**$_{\pm0.002}$ | 0.350 | 0.338 | 0.338 | 0.344 | 0.326 | 0.326 | 0.333 | 0.328 |
| ECL | 96 | **0.220**$_{\pm0.002}$ | 0.231 | 0.229 | 0.250 | 0.241 | 0.238 | 0.235 | 0.288 | 0.269 |
| | 192 | **0.238**$_{\pm0.001}$ | 0.254 | 0.250 | 0.263 | 0.255 | 0.251 | 0.250 | 0.295 | 0.273 |
| | 336 | **0.255**$_{\pm0.002}$ | 0.284 | 0.276 | 0.277 | 0.273 | 0.266 | 0.268 | 0.303 | 0.290 |
| | 720 | **0.288**$_{\pm0.003}$ | 0.346 | 0.337 | 0.309 | 0.313 | 0.297 | 0.293 | 0.325 | 0.322 |
| | Avg | **0.250**$_{\pm0.002}$ | 0.279 | 0.273 | 0.275 | 0.271 | 0.263 | 0.262 | 0.303 | 0.289 |
| Weather | 96 | **0.201**$_{\pm0.001}$ | 0.238 | 0.235 | 0.217 | 0.211 | 0.212 | 0.202 | 0.228 | 0.218 |
| | 192 | **0.248**$_{\pm0.001}$ | 0.290 | 0.285 | 0.259 | 0.251 | **0.248** | 0.250 | 0.269 | 0.259 |
| | 336 | 0.290$_{\pm0.002}$ | 0.336 | 0.327 | 0.297 | 0.291 | **0.286** | 0.292 | 0.310 | 0.297 |
| | 720 | **0.337**$_{\pm0.002}$ | 0.396 | 0.378 | 0.354 | 0.340 | 0.337 | 0.341 | 0.364 | 0.347 |
| | Avg | **0.269**$_{\pm0.002}$ | 0.315 | 0.306 | 0.282 | 0.273 | 0.271 | 0.271 | 0.293 | 0.280 |
| Traffic | 96 | **0.251**$_{\pm0.001}$ | 0.337 | 0.326 | 0.358 | 0.349 | 0.282 | 0.255 | 0.315 | 0.295 |
| | 192 | **0.259**$_{\pm0.002}$ | 0.323 | 0.315 | 0.376 | 0.362 | 0.290 | 0.268 | 0.317 | 0.296 |
| | 336 | **0.269**$_{\pm0.001}$ | 0.328 | 0.322 | 0.380 | 0.377 | 0.294 | 0.275 | 0.319 | 0.304 |
| | 720 | **0.284**$_{\pm0.004}$ | 0.351 | 0.344 | 0.394 | 0.391 | 0.312 | 0.297 | 0.335 | 0.322 |
| | Avg | **0.266**$_{\pm0.003}$ | 0.335 | 0.327 | 0.377 | 0.370 | 0.295 | 0.274 | 0.322 | 0.304 |

Table 4: Ablation analysis (MSE) on the ETTh1 and ETTh2 datasets. Lower values indicate better performance.

| Variant | ETTh1 | | | | ETTh2 | | | |
|---|---|---|---|---|---|---|---|---|
| | 96 | 192 | 336 | 720 | 96 | 192 | 336 | 720 |
| w/o TS modal | 0.358 | 0.389 | 0.402 | 0.425 | 0.276 | 0.332 | 0.359 | 0.382 |
| w/o VS modal | 0.355 | 0.389 | 0.412 | 0.441 | 0.273 | 0.336 | 0.359 | 0.386 |
| w/o projection | 0.353 | 0.385 | 0.405 | 0.416 | **0.266** | **0.328** | 0.360 | 0.376 |
| w/ all | **0.348** | **0.383** | **0.398** | **0.401** | **0.266** | **0.328** | **0.357** | **0.372** |

forecasting length of 720 is only increased by 0.75% and 5% compared to the forecasting length of 336 for the ETTh1 and ETTh2 datasets, respectively, showing competitive long-term forecasting capability.

### 4.2.1 Module Contribution Analysis

To analyze the contribution of each module to the predictive performance of VFEM, we conducted ablation experiments as shown in Table 4, which examine the impact of each modality processing module and the projection layer. From the results, the absence of any module leads to a decrease in performance, among which the visual module has a relatively greater impact on long-term forecasting. The projection layer also helps align information from the two modalities, facilitating cross-modal fusion.

Table 5: Ablation study on cross-modal fusion strategies on ETTh1. Best results are in **bold**.

| Fusion Type | $F$=96 | | $F$=192 | | $F$=336 | | $F$=720 | |
|---|---|---|---|---|---|---|---|---|
| | MSE | MAE | MSE | MAE | MSE | MAE | MSE | MAE |
| Add | 0.365 | 0.394 | 0.392 | 0.409 | 0.410 | 0.418 | 0.406 | 0.436 |
| Gated Add | 0.355 | 0.390 | 0.390 | 0.411 | 0.413 | 0.423 | 0.431 | 0.455 |
| Bilinear | 0.562 | 0.514 | 0.395 | 0.413 | 0.416 | 0.427 | 0.436 | 0.454 |
| Concat+MLP | 0.362 | 0.394 | 0.393 | 0.415 | 0.415 | 0.428 | 0.428 | 0.450 |
| Cross-Attn (TS→VS) | 0.357 | 0.388 | 0.388 | 0.407 | 0.424 | 0.434 | 0.438 | 0.463 |
| Cross-Attn (VS→TS) | 0.362 | 0.393 | 0.390 | 0.408 | 0.421 | 0.436 | 0.410 | 0.442 |
| Ours | **0.348** | **0.383** | **0.383** | **0.405** | **0.398** | **0.414** | **0.401** | **0.434** |

### 4.2.2 Cross-Modal Fusion Strategies

We compare several fusion strategies: (1) Add: element-wise addition; (2) Gated Add: addition with learnable gating weights; (3) Bilinear: bilinear pooling; (4) Concat+MLP: concatenation followed by MLP; (5-6) Cross-Attn: cross-attention with either temporal or visual features as query.

As shown in Table 5, our default strategy achieves the best performance across different prediction horizons. Simple addition performs competitively for long-term forecasting ($F$=720), while bilinear fusion shows instability at shorter horizons. Cross-attention variants show mixed results. In comparison, our concatenation-based approach with self-attention provides a simple and effective fusion.

### 4.2.3 Visual Encoder Freezing Strategies

Table 6: Ablation study on visual encoder freezing strategies. Best results are in **bold**.

| Strategy | ETTh1 | | | | ETTh2 | | | |
|---|---|---|---|---|---|---|---|---|
| | $F$=96 | | $F$=720 | | $F$=96 | | $F$=720 | |
| | MSE | MAE | MSE | MAE | MSE | MAE | MSE | MAE |
| Unfreeze-2 | 0.355 | 0.385 | 0.413 | 0.442 | 0.273 | 0.336 | 0.374 | 0.415 |
| Unfreeze-4 | 0.354 | 0.385 | 0.411 | 0.440 | 0.271 | 0.335 | 0.378 | 0.417 |
| Full Fine-tune | 0.353 | 0.384 | 0.423 | 0.452 | 0.272 | 0.336 | 0.378 | 0.417 |
| Ours | **0.348** | **0.383** | **0.401** | **0.434** | **0.266** | **0.330** | **0.372** | **0.413** |

We compare four freezing strategies: (1) Unfreeze-2: unfreeze the last 2 layers; (2) Unfreeze-4: unfreeze the last 4 layers; (3) Full Fine-tune: make all layers trainable; (4) Ours: keep all layers frozen.

As shown in Table 6, fully freezing the encoder achieves the lowest error. This is consistent with observations in cross-modal learning that freezing the pretrained encoder can help preserve its learned representations. When training with heterogeneous modalities, the gradient dynamics differ between branches, and unfreezing the visual encoder may affect its feature extraction capability.

### 4.2.4 Representation Quality Analysis

To validate that the projection layers learn meaningful time series representations, we compare three embedding types on ETTh1: (a) hand-crafted statistical features including mean, standard deviation, max, min, skewness, kurtosis, trend, and autocorrelation for each variable; (b) frozen SigLIP2 encoder outputs without fine-tuning; and (c) outputs after passing through the trained projection layers. We sample 500 sequences from training and test sets respectively, apply K-Means clustering ($k$=10), and evaluate using the Silhouette Score (Rousseeuw, 1987), which measures clustering quality by comparing intra-cluster cohesion with inter-cluster separation.

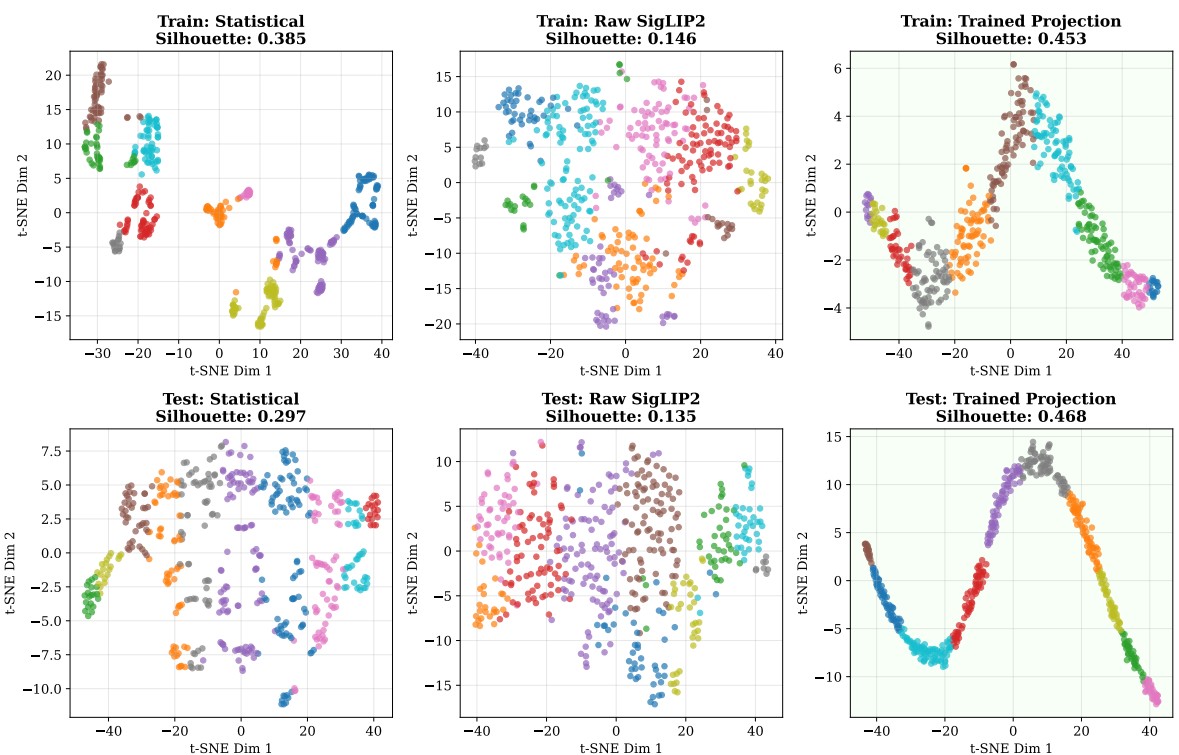

Figure 4: Embedding quality comparison on train and test sets. t-SNE visualizations of three embedding types: statistical features (left), raw SigLIP2 (center), and trained projection (right), evaluated on training set (top row) and test set (bottom row). Colors indicate K-Means cluster assignments ($k$=10). Silhouette Scores are shown for each configuration. The trained projection achieves the highest scores on both splits (0.453/0.468), demonstrating that the projection layers successfully adapt pre-trained visual features to the time series domain with strong generalization.

As shown in Figure 4, the trained projection achieves higher Silhouette Scores on both training set 0.453 and test set 0.468, compared to frozen encoder outputs at 0.146 and 0.135. The test set score slightly exceeds training with a gap of only 0.014, indicating good generalization. The low score of frozen SigLIP2 embeddings reflects the domain gap between natural images and time series, while the improvement after projection suggests that the projection layers help adapt visual features to the time series domain. The trained projection also outperforms statistical features with 0.468 versus 0.297 on test set, suggesting that the vision encoder captures patterns beyond hand-crafted statistics.

## 5    Conclusions

This work proposes VFEM, a cross-modal forecasting framework that integrates visual and temporal modalities for multivariate time series forecasting. By transforming time series into visual representations, VFEM enables pretrained vision models to perceive cross-variable spatial patterns that are difficult to capture with channel-independent architectures. The dual-branch design processes visual and temporal information independently before fusing them through self-attention.

By freezing the vision encoder and training only 7.45% of total parameters, VFEM leverages pretrained visual knowledge while avoiding overfitting on limited time series data. Ablation studies and representation analysis validate the effectiveness of each component. Experiments on seven benchmark datasets demonstrate that this cross-modal paradigm offers a viable alternative to existing approaches for capturing multivariate dependencies.

## Acknowledgments

This work was supported by Ant Group, the Shenzhen Ubiquitous Data Enabling Key Lab under Grant ZDSYS20220527171406015, the Tsinghua Shenzhen International Graduate School-Shenzhen Pengrui Endowed Professorship Scheme of Shenzhen Pengrui Foundation, and the Guangdong Basic and Applied Basic Research Foundation under Grant 2026A1515010184.

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

## A  Additional Spectral-Input Experiments

### A.1  Experimental Results and Analysis

To further examine whether spectral information can provide complementary evidence for forecasting, we evaluate five spectral representations: Continuous Wavelet Transform (CWT), Fast Fourier Transform (FFT), Mel Spectrogram, Short-Time Fourier Transform (STFT), and Wavelet Packet Transform (WPT). Detailed descriptions of each transformation are provided in Section A.2, with representative visualizations shown in Figure 5. In these experiments, the original time-series branch is retained, while one spectral representation replaces the raw time-series image at the visual branch and is fused with the time-series representation for forecasting. This allows us to compare different visual input representations within the same overall forecasting framework.

Table 7: Supplementary forecasting results on ETTh1 with alternative spectral inputs. Each setting retains the original time-series branch and incorporates a single spectral branch. Lower values indicate better performance.

| Method | Prediction Length | | | | | | | |
| | 96 | | 192 | | 336 | | 720 | |
| | MSE | MAE | MSE | MAE | MSE | MAE | MSE | MAE |
|---|---|---|---|---|---|---|---|---|
| Mel | 0.354 | 0.386 | 0.391 | 0.408 | 0.407 | 0.422 | 0.446 | 0.469 |
| WPT | 0.354 | 0.385 | 0.393 | 0.410 | 0.405 | 0.419 | 0.447 | 0.468 |
| STFT | 0.353 | 0.385 | 0.390 | 0.407 | 0.407 | 0.422 | 0.442 | 0.461 |
| CWT | 0.354 | 0.386 | 0.390 | 0.407 | 0.406 | 0.421 | 0.443 | 0.465 |
| FFT | 0.353 | 0.385 | 0.394 | 0.411 | 0.407 | 0.422 | 0.445 | 0.468 |
| Ours | **0.348** | **0.383** | **0.383** | **0.405** | **0.398** | **0.414** | **0.401** | **0.434** |

As shown in Table 7, the five spectral representations yield comparable forecasting accuracy on ETTh1, but none of them improves upon using raw multivariate time-series images as visual inputs. This may be because the visual backbone, pretrained on natural images, is better aligned with the spatial and shape patterns in raw time-series images than with the spectral patterns produced by these transformations. Further improvements may require spectral-domain pretraining or explicit cross-domain alignment, suggesting a possible direction for future research.

## A.2    Spectral Input Construction

Let the raw multivariate input be $\mathbf{X} \in \mathbb{R}^{B \times M \times L}$, where $B$ is the batch size, $M$ is the number of variables, and $L$ is the input sequence length. For the ETTh1 setup used in this supplementary study, $L = 512$ and $M = 7$.

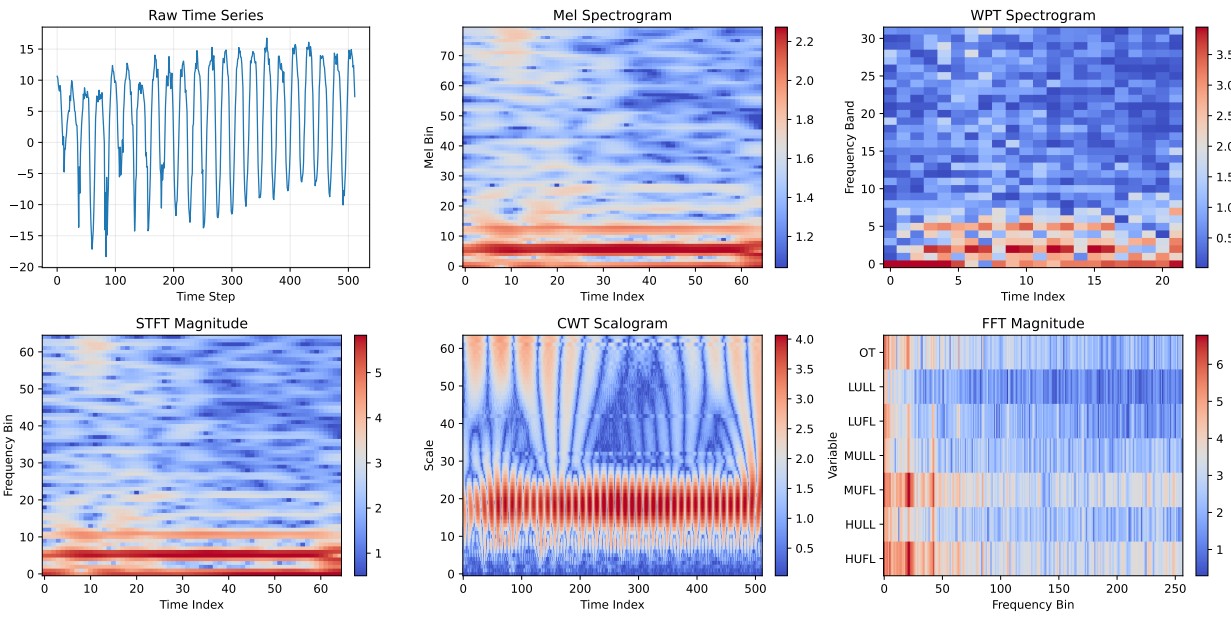

Figure 5: Visualization of an ETTh1 time-series sample and its spectral transformations. Mel, WPT, STFT, and CWT show different time-frequency or time-scale transformations of a single variable, while FFT shows a multivariate frequency-domain map where each row corresponds to the magnitude spectrum of one variable computed via per-variable FFT.

**Continuous Wavelet Transform.**    CWT maps each one-dimensional variable sequence into a time-scale representation. Using $S$ wavelet scales, the transformed input has shape $\mathbf{X}_{\mathrm{cwt}} \in \mathbb{R}^{B \times M \times S \times L}$. Following the same RGB expansion strategy, the batch and variable dimensions are merged, and the map is reshaped as $\mathbb{R}^{(B \times M) \times S \times L \times c}$ with $c = 3$ for vision encoding. The Morlet wavelet is adopted with $S = 64$. The resulting CWT representation preserves temporal location while describing variations across scales.

**Fast Fourier Transform.**    FFT produces a global frequency-domain magnitude representation. For a real-valued sequence of length $L$, the one-sided FFT has $F = L/2 + 1$ frequency bins. With $L = 512$, this gives $F = 257$, and the transformed representation is $\mathbf{X}_{\mathrm{fft}} \in \mathbb{R}^{B \times M \times F}$. Unlike time-frequency methods, FFT does not preserve temporal location. We arrange the global frequency magnitude along the variable and frequency dimensions, resulting in shape $\mathbb{R}^{B \times M \times F \times c}$.

**Mel Spectrogram.**    The Mel Spectrogram applies a short-time Fourier transform followed by a Mel filter bank and logarithmic compression, where $F$ denotes the number of Mel bins and $T$ the number of time frames, giving $\mathbf{X}_{\mathrm{mel}} \in \mathbb{R}^{B \times M \times F \times T}$. With $n_{\mathrm{fft}} = 128$, hop length 8, and $F = 80$ Mel bins, we have $T = 65$ for $L = 512$. After expansion, the Mel representation has shape $\mathbb{R}^{(B \times M) \times F \times T \times c}$.

**Short-Time Fourier Transform.**    STFT applies Fourier analysis over sliding windows, preserving both frequency and local temporal structure, and gives $\mathbf{X}_{\mathrm{stft}} \in \mathbb{R}^{B \times M \times F \times T}$. With $n_{\mathrm{fft}} = 128$ and hop length 8, the number of frequency bins is $F = 65$, and the number of time frames is approximately $T = 65$ for $L = 512$. After expansion, the STFT representation has shape $\mathbb{R}^{(B \times M) \times F \times T \times c}$.

**Wavelet Packet Transform.** WPT decomposes a sequence into wavelet packet coefficients across multiple frequency bands. With decomposition level $J$, the number of frequency bands is $2^J$, giving $\mathbf{X}_{\text{wpt}} \in \mathbb{R}^{B \times M \times 2^J \times T}$. The Daubechies-4 wavelet is used with $J = 5$, which gives 32 frequency bands and approximately $T = 22$ temporal positions for the input length considered here. After expansion, the WPT representation has shape $\mathbb{R}^{(B \times M) \times 2^J \times T \times c}$.

