# OpenReview forum: "VFEM: Visual Feature Empowered Multivariate Time Series Forecasting with Cross-Modal Fusion"
_TMLR — Accepted by TMLR_

### Review · Reviewer_8se7 · 2026-03-06

**Summary Of Contributions:**

The manuscript introduces VFEM, a cross-modal forecasting model designed to capture complex cross-variable patterns in multivariate time series by leveraging pre-trained large vision models (LVMs). The authors argue that while existing time series foundation models often use channel-independent architectures that ignore cross-channel dependencies , and current cross-modal methods rely heavily on text , rendering time series as 2D images allows LVMs to recognize spatial relationships. The model utilizes a dual-branch architecture consisting of a visual modality branch and a temporal modality branch , which are then fused using cross-modal attention.

**Audience:**

Yes

**Audience Explanation:**

- Novel Cross-Modal Application: Bypassing standard text-based methods to use Large Vision Models (LVMs) for spatial pattern recognition in time series is a fresh, creative approach.

- Parameter-Efficient Training: Freezing the vision encoder so that only 7.45% (30M) of the parameters require training is highly practical and appealing for compute-constrained research.

- Addressing Known Bottlenecks: The model explicitly tackles the failure of current channel-independent foundation models to capture complex cross-variable dependencies.

- Rigorous Validation: The evaluation across 7 standard benchmarks and the robust representation analysis using t-SNE and Silhouette scores provide exactly the kind of concrete empirical evidence TMLR readers expect

**Claims And Evidence:**

Yes

**Claims Explanation:**

- The authors conducted extensive experiments across 7 distinct datasets (ETTh1, ETTh2, ETTm1, ETTm2, Electricity, Weather, and Traffic). They benchmarked VFEM against a strong and diverse set of baselines, including foundation models like Chronos and Moirai, cross-modal models like GPT4TS, and specialized architectures like PatchTST. The results in Table 2 clearly demonstrate that VFEM consistently achieves lower MSE and MAE metrics across multiple forecasting horizons.

- The paper includes a thorough ablation study (Table 3) on the ETTh1 and ETTh2 datasets.The study explicitly shows that removing the visual modality, the temporal modality, or the projection layer results in increased error rates. The authors also provide a solid comparison of different cross-modal fusion strategies (Table 4), proving their chosen concatenation and self-attention approach yields the best results.

- The authors detail their parameter distribution in Table 1, showing that by freezing the SigLIP-2 vision encoder, only 30M parameters (7.45% of the total) require training. Furthermore, they support their decision to freeze the encoder with an ablation study (Table 5) that proves fully freezing the vision encoder actually produces lower forecasting errors than unfreezing its layers or performing full fine-tuning.

- The submission uses t-SNE visualizations and Silhouette scores (Figure 4) to evaluate embedding quality. The analysis proves that the trained projection layers successfully adapt the visual features to the time series domain. This is a measurable improvement over both raw, frozen LVM embeddings and hand-crafted statistical features.

- However, there are a couple of minor areas where the evidence could be stronger:
1. Missing Baseline Comparisons: The authors note that the Chronos and Moirai baseline models did not report results on the Traffic dataset because it possesses an excessive number of variables. While this highlights VFEM's scalability, it leaves a gap in direct comparative evidence against large time series models on highly multivariate datasets.
2. Global vs. Local Context: The model design extracts a single global visual embedding and replicates it across all variables. The paper does not provide empirical evidence comparing this global replication strategy against extracting variable-specific (local) visual features.

**Requested Changes:**

While the paper presents a strong methodology, there're some further improvements that can be done:

1. Global vs. Local Visual Embeddings: The visual branch processes the input to obtain a single global image representation. This global embedding is then replicated across all $M$ variables using variable-wise expansion so each variable can access the shared context during fusion. The authors might consider discussing the limitations of relying purely on a global embedding versus extracting variable-specific local visual features.

2. Baseline Missing Data: The authors state that the Chronos and Moirai baselines did not report results on the Traffic dataset due to its excessive number of variables. While it is a strength that VFEM can handle this dataset, it leaves a gap in direct comparison against these specific large-scale foundation models in highly multivariate scenarios.

3. Clustering Validation: The t-SNE visualization and Silhouette Score analysis effectively demonstrate that the trained projection layers adapt visual features to the time series domain. However, evaluating this on datasets beyond ETTh1 could further strengthen claims of broad domain generalization.

---

> ### Author Response · Authors · 2026-04-05
>
> Thank you for your careful review of our paper. Your comments are beneficial to further improving this work. We address your questions one by one as follows.
>
> **Q1. On global and local visual embeddings, comparison, and limitations**
>
> We agree with the reviewer's insight regarding the visual embedding branch. In fact, we have carried out substantial exploration and experimentation on visual representation conversion.
>
> Specifically, the current visual embedding process can be summarized as follows. For the visual input $X \in \mathbb{R}^{M \times L \times c}$, where $M$ is the variable dimension, $L$ is the temporal dimension, and $c = 3$ is the RGB channel dimension of the image, the visual encoder outputs $e_{vs} \in \mathbb{R}^{d_v}$. This one-dimensional representation captures global visual information. It is then transformed through variable-wise expansion into $E_{vs} \in \mathbb{R}^{M \times d_v}$, so that each channel in the visual branch contains the global visual embedding. After that, a feature projection layer produces the channel-level global representation $H_{vs} \in \mathbb{R}^{M \times d}$, where $d$ is the projected representation dimension, in order to adapt to the characteristics of each variable and the needs of downstream cross-modal fusion. In this process, every channel in the visual branch contains a channel-adapted version of the global visual embedding.
>
> Overall, this process involves three important dimensional transformations:
>
> $$X \in \mathbb{R}^{M \times L \times c} \rightarrow e_{vs} \in \mathbb{R}^{d_v} \rightarrow E_{vs} \in \mathbb{R}^{M \times d_v} \rightarrow H_{vs} \in \mathbb{R}^{M \times d}.$$
>
> For the first transformation, we also tried converting $X$ into
>
> $X \in \mathbb{R}^{M \times P \times N \times c}, \quad P \times N = L,$
>
> where $P$ is the patch length and $N$ is the number of patches. Feeding this into the visual encoder gives
>
> $e_{vs} \in \mathbb{R}^{M \times d_v},$
>
> where $M$ is the number of variables, thereby constructing a univariate time-series image representation. Similarly, if we swap the dimensions $N$ and $M$, we obtain
>
> $X \in \mathbb{R}^{N \times P \times M \times c},$
>
> which yields
>
> $e_{vs} \in \mathbb{R}^{N \times d_v}$
>
> after the visual encoder, thereby constructing a temporally local spatiotemporal visual representation.
>
> The representations obtained from these two local visual strategies can be directly used as the two-dimensional representation corresponding to the second transformation in the global-representation method, i.e., without the need for variable-wise expansion from one-dimensional to two-dimensional representation count. Then, in the projection layer, the first local strategy projects along $d_v$, while the second local strategy needs to project both $N$ and $d_v$ so that it can match the temporal branch and adapt to cross-modal fusion.
>
> We tested both of these transformation mechanisms, but neither performed better than the current global representation. Our current understanding is that explicitly using cross-channel information and long-range temporal information in the visual representation better exploits the ability of the visual encoder, because modern visual encoders already have the capacity to represent both global and local information.
>
> **Q2. Missing Traffic baseline data**
>
> We agree that the lack of Chronos and Moirai results on the Traffic dataset makes the comparison less sufficient. The Traffic dataset contains 861 variables, which makes channel-wise computation difficult. As suggested in [1], it would be better to consider more reliable benchmarks when evaluating forecasting performance. The Traffic dataset also has multiple variants (e.g., PeMS03, PeMS04, PeMS07, PeMS08). We will try to supplement Chronos and Moirai baseline results on Traffic, or alternatively replace it with comparisons on multiple Traffic dataset variants.
>
> [1] Brigato, Lorenzo, et al. Position: There Are No Champions in Long-Term Time Series Forecasting. TMLR 2025.
>
> **Q3. Generalization of clustering-based validation for visual representation adaptation to the time-series domain**
>
> We agree that further t-SNE visualization and Silhouette Score analysis on more datasets would help strengthen the claim that the visual projection layer adapts visual features to the time-series domain. We will consider adding visualizations on time-series data from other domains in the revised version for further validation.

---

### Review · Reviewer_Zoz5 · 2026-03-08

**Summary Of Contributions:**

In this paper, the authors study the problem of multivariate time-series forecasting. To this end, they propose a multimodal architecture wherein a visual transformation of time-series data is processed by a large vision model (LVM) and the raw timeseries data is processed by a temporal model. The representations from the two branches are fused in a simple manner and mapped to the target values. Across different, standard datasets, the proposed approach is shown to provide state-of-the-art results.

Strengths:
+ Multivariate time-series forecasting is an important problem.
+ Significant improvements are reported across different datasets.
+ The text is generally easy to follow, except for the Introduction.

**Audience:**

Yes

**Audience Explanation:**

Multi-variate timeseries forecasting is an important problem in ML.

**Broader Impact Concerns:**

Not applicable.

**Claims And Evidence:**

No

**Claims Explanation:**

2. Introduction criticizes prior work for being channel independent. However:

2.1. Even after reading the paper, it is still not clear to me what is meant by a channel-(in)dependent architecture. If it pertains to the RGB channels of the constructed image, I fail to follow how this is significant, considering the fact that RGB channels are all replicas of the same visual content.

2.2. Introduction: "Firstly, the channel-independent architecture is commonly adopted in large time series models" => It is not clear at this point what a/the channel-independent architecture is.

2.3. A teaser figure providing an overview summary of how the proposed approach differs from earlier work would be helpful in Introduction to be able to follow the terminology and the contributions.

2.4. There should be a systematic comparison with a channel-independent model (or a version of your model) to justify this claim.

**Requested Changes:**

Weaknesses:

1. The paper is a very simple and straightforward fusion of an LVM and a temporal model. I do not see sufficient technical novelty for the TMLR journal.

2. Introduction criticizes prior work for being channel independent. However:

2.1. Even after reading the paper, it is still not clear to me what is meant by a channel-(in)dependent architecture. If it pertains to the RGB channels of the constructed image, I fail to follow how this is significant, considering the fact that RGB channels are all replicas of the same visual content.

2.2. Introduction: "Firstly, the channel-independent architecture is commonly adopted in large time series models" => It is not clear at this point what a/the channel-independent architecture is.

2.3. A teaser figure providing an overview summary of how the proposed approach differs from earlier work would be helpful in Introduction to be able to follow the terminology and the contributions.

2.4. There should be a systematic comparison with a channel-independent model (or a version of your model) to justify this claim.

3. The compared methods include mainly Transformer-based approaches. I would suggest including comparisons with Mamba-based methods, e.g.:

ms-mamba: Multi-scale mamba for time-series forecasting, Neurocomputing, 2026.


Minor comments:
- " finance Allen et al. (2025); Wu et al. (2021b); Liu et al. (2023); Yi et al. (2023)" => In such cases, references should be within parentheses.

- Figure 1 should be positioned closer to its first reference on page 1.

- Eq 1: Please introduce lowercase \hat{y} and y.

- Figure 2: Please increase the contrast for the white text in green/purple boxes.

---

> ### Author Response · Authors · 2026-04-05
>
> We sincerely appreciate your time and effort in carefully reviewing our manuscript. Your valuable advice will help us refine this work. Below are our responses to your questions.
>
> **Q1. On the meaning of channel-(in)dependent architecture and its relation to image RGB channels**
>
> Thank you for raising this question. We agree that the manuscript should include a necessary explanation of channel-independent and channel-dependent structures. Briefly, as described in the problem formulation subsection of our paper, the input time series is denoted as $X \in \mathbb{R}^{L \times M}$, where $L$ is the sequence length and $M$ is the number of variables. The dimension $M$ here represents the channel dimension, and in machine learning the terms "variable" and "channel" are often used interchangeably [1][2].
>
> More specifically, in the PatchTST paper [3], the proposed channel-independent architecture processes each univariate time series through patching and applies the attention mechanism to each univariate series, transforming the length dimension $L$ into $P \times N$, where $P$ is the patch length and $N$ is the number of patches. By contrast, a channel-dependent structure explicitly considers the dependencies among different variables. For example, the routing mechanism in Crossformer [4] captures dependencies among different time-series variables.
>
> Since our work involves both the time-series modality and the image modality, it is easy to confuse the channel concept in time series with the RGB channel dimension $c$ in the image input $X_{vs} \in \mathbb{R}^{M \times L \times c}$. We will clarify in the revision that these are concepts from different modalities.
>
> [1] Wang, Muyao, Zeke Xie, Bo Chen, Hongwei Liu, and James Kwok. Channel Matters: Estimating Channel Influence for Multivariate Time Series. NeurIPS 2025.
>
> [2] Zhao, Lifan, and Yanyan Shen. Rethinking Channel Dependence for Multivariate Time Series Forecasting: Learning from Leading Indicators. ICLR 2024.
>
> [3] Nie, Yuqi, Nam H. Nguyen, Phanwadee Sinthong, and Jayant Kalagnanam. A Time Series Is Worth 64 Words: Long-Term Forecasting with Transformers. ICLR 2023.
>
> [4] Zhang, Yunhao, and Junchi Yan. Crossformer: Transformer Utilizing Cross-Dimension Dependency for Multivariate Time Series Forecasting. ICLR 2023.
>
> **Q2. On adding an overview schematic in the references/related-work part and supplementing comparison with channel-independent models**
>
> We agree that adding an overview schematic or table would make the comparison with related work clearer and more reader-friendly. We will seriously consider how to improve this part in the revised version.
>
> Regarding comparison with channel-independent models, we have already used multiple channel-independent models as baselines, including Chronos [5], GPT4TS [6], PatchTST, and TimesNet [7]. We also used channel-dependent model structures as baselines, including Moirai [8] and UniTST [9]. However, as discussed in our related-work section, our model has both a channel-dependent temporal branch and an image-modality branch, and then performs cross-modal fusion of the temporal and image representations in the backbone. In this sense, our model is significantly different from previous work. It provides a new perspective for cross-channel and cross-modal modeling of multivariate time series, and shows that visual enhancement can bring performance gains while requiring only a small-model-scale amount of parameter training, with potential advantages for longer-horizon forecasting.
>
> [5] Abdul Fatir Ansari, Lorenzo Stella, Ali Caner Turkmen, et al. Chronos: Learning the Language of Time Series. TMLR 2024.
>
> [6] Tian Zhou, PeiSong Niu, Xue Wang, Liang Sun, Rong Jin. One Fits All: Power General Time Series Analysis by Pretrained LM. NeurIPS 2023.
>
> [7] Haixu Wu, Tengge Hu, Yong Liu, Hang Zhou, Jianmin Wang, Mingsheng Long. TimesNet: Temporal 2D-Variation Modeling for General Time Series Analysis. ICLR 2023.
>
> [8] Gerald Woo, Chenghao Liu, Akshat Kumar, Caiming Xiong, Silvio Savarese, Doyen Sahoo. Unified Training of Universal Time Series Forecasting Transformers. ICML 2024.
>
> [9] Juncheng Liu et al. UniTST: Effectively Modeling Inter-Series and Intra-Series Dependencies for Multivariate Time Series Forecasting. TMLR 2025.

---

> > ### Author Response · Authors · 2026-04-05
> >
> > **Q3. On adding Mamba-based models as comparison baselines**
> >
> > We agree that adding comparisons with Mamba-based models would enrich the structural diversity of the baselines. That said, the main goal of the current work is to explore visual enhancement for cross-channel and cross-modal modeling, and the current experimental setting is self-consistent. Still, we agree that this rapidly developing family of models should be included [10][11][12]. We will seriously consider including more model structures, including Mamba-based time-series models, in the related-work discussion and analysis. Thank you very much for this valuable suggestion.
> >
> > [10] Zihan Wang, Fanheng Kong, Shi Feng, Ming Wang, Xiaocui Yang, Han Zhao, Daling Wang, Yifei Zhang. Is Mamba Effective for Time Series Forecasting? Neurocomputing, 2024.
> >
> > [11] Yoo-Min Jung and Leekyung Kim. MambaSL: Exploring Single-Layer Mamba for Time Series Classification. ICLR 2026.
> >
> > [12] Yusuf Meric Karadag, Ismail Talaz, Ipek Gursel Dino, Sinan Kalkan. ms-Mamba: Multi-scale Mamba for Time-Series Forecasting. Neurocomputing, 2026.

---

### Review · Reviewer_j2jm · 2026-03-16

**Summary Of Contributions:**

The paper introduces a novel neural architecture for multivariate time series forecasting that treats the input as two complementary modalities: the raw time series and an image. The original time series is processed by a temporal attention block, while the image. constructed as an intensity map with axes corresponding to features and time steps, is processed by a pre-trained, frozen vision transformer. The resulting embeddings are fused to produce the final forecast.

The proposed architecture demonstrates competitive performance on standard long-horizon time series benchmarks covering weather and energy domains, reportedly outperforming both univariate time series foundation models (Chronos, Moirai) and specialized supervised architectures (TimesNet, PatchTST) on MSE and MAE across multiple prediction horizons. The authors support these claims with ablation experiments on the fusion strategy and provide qualitative intuition for the contribution of the vision modality by illustrating the patterns visible in the intensity maps.

**Audience:**

Yes

**Audience Explanation:**

Adding vision modality processing is a technically simple extension to pretty much any underlying time series backbone. Since the vision transformer is frozen, the computational overhead is relatively small. This presents a practical and straightforward way to improve performance of time series models, which would be a useful contribution to the field.

**Broader Impact Concerns:**

No concerns (new method on standard benchmark datasets).

**Claims And Evidence:**

No

**Claims Explanation:**

The paper states that *"for fair comparison, we adopt the results reported in each model's original publication, ensuring that all baselines operate under their optimal configurations"* (Section 4.1, 3rd paragraph). However, the **Chronos paper reports WQL and MASE, not MSE and MAE** (Ansari et al., 2024, https://arxiv.org/pdf/2403.07815). The Chronos MSE/MAE numbers in the comparison table therefore cannot have come from the original publication.

To the best of my knowledge, **these values appear to originate from the Time-MoE paper** (Table 3 in Shi et al., 2024b; https://arxiv.org/pdf/2409.16040 ; which is cited by the Sundial paper: Table 9 in Liu et al., 2025; https://arxiv.org/pdf/2502.00816), which reports Chronos as a baseline and is also **not cited in this context**. This directly contradicts the authors' stated evaluation protocol and raises questions about the provenance of the reported numbers. Results for Times-MoE and Sundial are not reported.

The results for **Chronos and Moirai are obtained in a zero-shot mode, not fine-tuned**, while the proposed method is fully supervised on the datasets. This crucial distinction is not made clear and the claim that the zero-shot is an "optimal configuration" for the time series model is not completely accurate.

Further, there are no ablations on (1) limitations of the proposed approach, (2)  the choice of the time and vision backbones, (3) the engineering of the image modality from time series data.

**Requested Changes:**

- __Major__
    - **Misattribution of the reported results**. Chronos paper does not report MSE and MAE, the numbers in the comparison table appear to originate from the Times-MOE paper (Table 3 in Shi et al., 2024b; https://arxiv.org/pdf/2409.16040), which is not cited.
	- **Unfair comparison with foundation model baselines**. Chronos and Moirai are used in zero-shot mode in their original publications, while the proposed method is fully supervised. This comparison should be clearly stated and justified.
- Results
	- **Recent baselines**. Times-MOE and Sundial should be cited and included as papers sharing the same evaluation protocol. Recent multivariate time-series models like Chronos-2 should probably also be included.
    - **Backbone ablation study**. The paper does not investigate whether the performance gains from the vision modality are specific to the chosen time series backbone or generalise across architectures. It would be informative to evaluate the vision extension with at least one alternative backbone (e.g., PatchTST or a foundation model in zero-shot or fine-tuning mode) to establish generality.
	- **Image modality engineering**. Only a single type of image representation (the intensity map) is explored. It is unclear whether the performance gain is due to the general principle of incorporating visual information, or specific properties of this particular construction. Comparing alternative representations (e.g.,  frequency-domain images) would help clarify what the vision model actually captures and whether the approach is robust to the choice of image construction
	- **Practical advice on feature and time steps number**. The paper does not characterise the conditions under which the vision modality is expected to help. For instance, how does performance scale with the number of input features, which determines the spatial resolution of the intensity map? What is the minimum number of time steps required for the image representation to be informative? Establishing these boundaries would significantly strengthen the practical contribution.
	- **Confidence intervals**. Experiments are run with three seeds but no confidence intervals or standard deviations are reported. These should be included.
	-  **Metrics**. Some papers (like Chronos) use a different set of metrics (WQL and MASE), so these should be also reported or there should be a more specific justification for choosing MSE and MAE.
- Reporting
 	- Table 3 Metric not indicated in Table 3
	- Table 3 is transposed relative to Table 2, making it hard to compare
    - Incorrect citation typing (citet vs citep), breaking the flow of text and making hard to read.
	- Colormaps are inconsistent across figures
	- Consider using the booktabs table style for improved readability
	- Table 2 reports both MSE and MAE, making it hard to parse; consider presenting separate tables

---

> ### Author Response · Authors · 2026-04-05
>
> First of all, we would like to thank you for taking the time to carefully review our paper. Your constructive suggestions are helpful for improving this work. We respond to your comments as follows.
>
> **Q1. On the provenance of baseline results and the training regime**
>
> We agree with your observation about the baseline setup for the large time-series foundation models. For the Chronos and Moirai baselines, since the original papers did not provide MSE/MAE results, the values we actually used came from the corresponding results reported in the Sundial paper [1], and may have originally traced back to the benchmark reporting in the Time-MoE paper [2]. We will clarify this baseline setup and add the relevant citations in the revised version.
>
> We also accept that our claim about "optimal configuration" was not rigorous enough, and we will clearly state that the baselines for these time-series foundation models came from the zero-shot setting. At the same time, we will also conduct fully supervised training for the foundation model baselines and compare with fine-tuned baselines.
>
> VFEM is a vision-enhanced cross-modal fusion model that does not require pretraining on large-scale time-series datasets. After freezing the visual encoder, the number of parameters that need to be fine-tuned is comparable to that of a small model. This allows VFEM to leverage the feature extraction capability of a pretrained vision encoder while only fine-tuning the temporal branch and fusion layers to obtain overall performance gains.
>
> **Q2. On adding Time-MoE and Sundial as baselines**
>
> We agree that Time-MoE and Sundial are both strong time-series foundation models, and that they each show strong forecasting performance in their reported zero-shot results. We do not claim that VFEM achieves the best performance in time-series forecasting; what we would like to emphasize is instead the new vision-enhancement mechanism in VFEM: even without large-scale time-series pretraining, the visual encoder helps improve the spatiotemporal modeling ability of multivariate time-series features.
>
> We view this as an essential direction for future work: studying whether large-scale time-series pretraining can be combined with visual-modality enhancement for further gains. In addition, as suggested in [3], different models are suitable for different application scenarios. With visual-modality enhancement, VFEM shows lower performance degradation at longer forecasting horizons, while requiring only small-model-scale parameter training.
>
> [1] Yang, Chengyue, Xinyu Huang, Anni Ren, Yujing Wang, and Ming Zhong. Sundial: A Family of Highly Capable Time Series Foundation Models. ICML 2025.
>
> [2] Shi, Xiaoming, Shiyu Wang, Yuqi Nie, Dianqi Li, Zhou Ye, Qingsong Wen, and Ming Jin. Time-MoE: Billion-Scale Time Series Foundation Models with Mixture of Experts. ICLR 2025.
>
> [3] Brigato, Lorenzo, et al. Position: There Are No Champions in Long-Term Time Series Forecasting. TMLR 2025.
>
> **Q3. On additional ablation experiments**
>
> Regarding the choice of branch structure and the conversion from time series to image modality, we did in fact try multiple designs. For the temporal branch, we explored MLP-based and attention-based structures. We also tried different visual encoder backbones, including ViT, DINOv2, BLIP2, and the non-NaFlex version of SigLIP2. However, these models do not support arbitrary aspect-ratio inputs.
>
> This matters in our setting because we use the entire multivariate time series as image-modality input. For datasets with a small number of variables, this requires padding, and it also requires adaptation along both the temporal and variable dimensions. The padding process may dilute useful information and increase computational cost. This is especially problematic for datasets where the number of variables and the sequence length are highly imbalanced, leading to performance degradation and increased training time.
>
> By contrast, the NaFlex version of SigLIP2, which supports arbitrary aspect ratios, only requires the conversion procedure shown in Figure 3 of the manuscript to form the image-modality input. In this sense, it better matches the input characteristics of multivariate time series datasets.

---

> > ### Author Response · Authors · 2026-04-05
> >
> > **Q4. On image-modality pattern validation, and evaluation metrics**
> >
> > In fact, we tried several frequency-domain image patterns as input, including converting a single time series into mel-spectrograms, short-time Fourier transform images, and wave packet transform images. However, their performance was not better than the current approach. One possible reason is that converting a single-variable time series into a frequency-domain image does not explicitly capture cross-channel information. Another possibility is that multivariate time-series intensity maps are more intuitive, whereas spectral images may require additional supervised adaptation of the visual encoder.
> >
> > As for evaluation metrics, we did not use WQL because the current model structure is designed for point forecasting rather than distribution forecasting. MASE adds a comparison against a naive model on top of MAE. We believe this metric is important toward broader evaluation standards; however, we are not in favor of including it in the current version, because the current error-based metrics already make the models comparable and are sufficient to support the main claim of our work.
> >
> > **Q5. On table format, confidence intervals, citation style, figure color design**
> >
> > Thank you very much for these specific and careful suggestions. We will seriously consider the following changes in the revised version: adding explicit metric labels to Table 3, adopting proper citation style, adjusting the table format and figure color design to improve readability, and reporting confidence intervals for the main experimental results based on more random seeds. We appreciate these detailed suggestions.

---

### Decision · Action_Editor_aBCh · 2026-04-23

**Recommendation:** Accept with minor revision

**Additional Comments:**

We request that the authors incorporate the revisions requested by the reviewers, especially the ones requested by reviewer j2jm.

**Audience:**

Yes

**Audience Explanation:**

The reviewers agree that this simple and efficient method could be interesting for some members of the TMLR audience.

**Claims And Evidence:**

Yes

**Claims Explanation:**

The reviewers find that the paper is a simple and straightforward fusion of existing approaches, but that its efficiency is laudable. A few key claims require adjustment, which the authors promised to revise, hence we request a minor revision at this point (see details below).

---

> ### Author Response · Authors · 2026-05-09
>
> Dear Action Editor and Reviewers,
>
> We sincerely thank you for your constructive feedback and decision on our manuscript. We have now submitted the camera-ready version. In this revision, we clarified the metric description in the ablation table, standardized citation formatting, improved table structure using the booktabs style, made additional minor formatting adjustments for consistency, and added a brief mention of recent progress in Mamba-based state-space architectures in the Introduction.
>
> We also split the main results into separate MSE and MAE tables for better readability, added standard deviation reporting for VFEM, supplemented Chronos and Moirai results on the Traffic dataset, and enhanced numerical readability through table typography and spacing adjustments. We sincerely appreciate your comments, which helped us improve the clarity and overall quality of the manuscript.
>
> Best regards,
>
> Authors

---

> > ### Comment · Reviewer_Zoz5 · 2026-05-10
> > **Re:**
> >
> > Dear all,
> >
> > I want to confirm that I have checked the final version of the paper and I am happy with the changes performed.
> >
> > Best

---

> > > ### Author Response · Authors · 2026-05-10
> > >
> > > Dear Reviewer Zoz5,
> > >
> > > Thank you for confirming the final version and your positive feedback.
> > >
> > > Best,
> > >
> > > Authors